# Altered expression of Notch1 in Alzheimer's disease

**Sun-Jung Cho**[1], **Sang-Moon Yun**[1], **Chulman Jo**[1], **Jihyun Jeong**[1], **Moon Ho Park**[2], **Changsu Han**[3], **Young Ho Koh**[1]*

**1** Division of Brain Diseases, Center for Biomedical Sciences, Korea National Institute of Health, 187 Osongsaengmyeong2-ro, Osong-eup, Heungdeok-gu, Cheongju-si, Chungcheongbuk-do, Republic of Korea, **2** Departments of Neurology, Korea University Medical College, Ansan Hospital, 123 Jeokgeum-ro, Danwon-gu, Ansan-si, Gyeonggi-do, Republic of Korea, **3** Departments of Psychiatry, Korea University Medical College, Ansan Hospital, 123 Jeokgeum-ro, Danwon-gu, Ansan-si, Gyeonggi-do, Republic of Korea

* kohyoungho122@gmail.com

## Abstract

Notch signaling is an evolutionarily conserved pathway that regulates cell-cell interactions through binding of Notch family receptors to their cognate ligands. Notch signaling has an essential role in vascular development and angiogenesis. Recent studies have reported that Notch may be implicated in Alzheimer's disease (AD) pathophysiology. We measured the levels of soluble Notch1 (sNotch1) in the plasma samples from 72 dementia patients (average age 75.1 y), 89 subjects with amnestic mild cognitive impairment (MCI) (average age 73.72 y), and 150 cognitively normal controls (average age 72.34 y). Plasma levels of sNotch1 were 25.27% lower in dementia patients as compared to healthy control subjects. However, the level of Notch1 protein was significantly increased in human brain microvascular endothelial cells (HBMECs) after amyloid-beta treatment. Also, *Notch1* mRNA level was significantly increased in HBMECs and iPSC-derived neuronal cells from AD patient compared to normal control. These results indicate that altered expression of Notch1 might be associated with the risk of Alzheimer's disease.

## Introduction

Alzheimer's disease (AD) is a complex neurodegenerative syndrome caused by abnormal accumulation of biological products for decades. Recently, it has been reported that Notch signaling in cerebrovascular diseases is associated with angiogenesis and the functioning of the blood-brain barrier (BBB)[1]. Notch1 could mediate neurodegenerative progress including AD. Notch is involved in regulating proteolytic processing of amyloid precursor protein (APP) through cleavage by ADAM10/17 and γ-secretase[2, 3]. The membrane-bound Notch receptor is cleaved into a secreted Notch extracellular truncation domain (NEXT) and an activated Notch intracellular domain (NICD). NICD translocates into the nucleus and modulates the expression of several genes[4]. Notch signaling cascade is involved in various intracellular signaling processes including BBB functioning, cerebrovascular formation, and the development of tissue-specific cell types[5, 6]. In BBB, during the vessel formation process, vascular

**Data Availability Statement:** We cannot publicly provide individual data due to participants' privacy specified by the Institutional Review Board (IRB) of the Korea Centers for Disease Control and Prevention (KCDC). In addition, the written

informed consent we obtained from study participants does not include a provision for publicly sharing data. Qualifying researchers may apply to access a minimal dataset by contacting the current data access coordinator, M.S. Sunhyo Kim via ssunyo88@korea.kr.

**Funding:** This work was supported by the Korea Centers for Disease Control and Prevention (KCDC) 2016-NG62002-00, 2017-NI62001-00, (http://www.cdc.go.kr). The funder had no role in study design, data collection and analysis, decision to publish, or preparation of the manuscript.

**Competing interests:** The authors have declared that no competing interests exist.

endothelial growth factor A (VEGFA) is the most potent mitogen acting on endothelial cells (ECs). VEGFA stimulates vasculogenesis through VEGF receptor 2 (VEGFR2)-induced activation of DLL4 and Notch1 signaling pathway[7]. A recent study showed that Notch and VEGF signaling mediates vasculogenesis and angiogenesis through regulation of VEGFR2 in BBB[8].

In the brain, Notch signaling plays an important role in the developmental processes, synaptic plasticity, and glial cell activation[9–11]. Notch might also be involved in the neural stem cell activation in response to exercise, learning, or injury. However, the mechanism of Notch signaling-mediated fine-tuning of neurogenesis is not fully understood. Recently, it has been reported that Notch1 protein expression is significantly increased in the brain of patients with AD and Notch1 is deposited in $A\beta_{1-42}$ positive plaques[12]. Plasma Notch1 and tumor necrosis factor-α converting enzyme (TACE) levels were significantly higher in the abdominal aortic aneurysm (AAA) postulating that the Notch1 signaling in macrophages plays an important role in AAA development and progression[13]. However, the plasma level of Notch1 has never been explored in AD. The level of full-length and truncated Notch1 in the cerebrospinal fluid (CSF) is reduced in AD patients suggesting its accumulation in the brain parenchyma[12]. These studies indicate that imbalance of Notch1 signaling might be implicated in AD. In our previous study, we examined the plasma levels of soluble VEGFR2 (sVEGFR2) are significantly decreased in AD patients[8]. We found that that the *NOTCH1* mRNA level was significantly increased in human umbilical vein endothelial cells (HUVECs) after Aβ treatment. Overexpression of NICD significantly decreased the *sVEGFR2* and *VEGFR2* mRNA levels in HUVEC indicating that the altered VEGFR2 expression might be associated with AD[8].

In this study, we evaluated the changes in the plasma levels of Notch1 in AD and determined the role of Notch1 in regulating cerebrovasculogenesis in ECs. We demonstrated that the soluble Notch1 (sNotch1) levels are significantly decreased in the plasma of AD patients. These results highlight the importance of Notch1 as a potent biomarker in patients at risk for AD.

## Materials and methods

### Subjects

Control, subjects with mild cognitive impairment (MCI) and dementia subjects were selected from the population-based Ansan Geriatric (AGE) cohort established in 2002 to study the common geriatric diseases of elderly Koreans aged between 60–84 years. The sampling protocol and design of the AGE study have been previously described[14, 15] and the population of this study has referenced in our previous study[8, 16]. Cognitive functioning and memory impairments of the subjects were assessed using a Korean version of Consortium to Establish a Registry for Alzheimer's disease (CERAD-K) neuropsychological battery[17]. The basic structures of all measures in the original CERAD batteries were maintained in Korean translation. All participants were clinically evaluated according to published guidelines. All the dementia patient met the criteria as described in the Diagnostic and Statistical Manual of Mental Disorders, fourth edition[18] and the National Institute of Neurological and Communicative Disorders and Stroke and the Alzheimer's Disease and Related Disorders Association (NINCDS–ADRDA)[15]. MCI was diagnosed on the basis of the Mayo Clinic criteria [19] as described previously[20, 21]. In total, blood samples from 311 subjects were included in our study. The distribution of control, MCI, and dementia subjects are shown in Table 1. The study subjects consisted of 72 dementia patients (average age 75.1 ± 0.68, 17 males, 55 females), 89 subjects with MCI (average age 73.72 ± 0.52, 36 males, 53 females), and 150 unrelated healthy control subjects (average age 72.34 ± 0.37, 65 males, 85 females). Table 1 summarizes demographic and clinical measures for all covariates tested here, including diagnosis (normal, MCI,

**Table 1. Baseline characteristics of the study population.**

| Features | Control | MCI | Dementia | *p*-value |
|---|---|---|---|---|
| N (Male/Female) | 150(65/85) | 89(36/53) | 72(17/55) | |
| Age (yr) | 72.34±0.37 | 73.72±0.52 | 75.1±0.68 | <0.001 |
| Education | 9.28±0.39 | 6.02±0.51 | 3.30±0.51 | < 0.001 |
| MMSE | 27.33±0.17 | 24.65±0.34 | 15.92±0.68 | <0.001 |
| CDR | 0.037±0.01 | 0.26±0.02 | 1.11±0.08 | <0.001 |
| sNotch1 (ng/mL) | 2.81±0.17 | 2.23±0.07 [a] | 2.1±0.07 [b] | **0.015** |

Values are mean ± SEM

[a]compared with control

[b]compared with control; $p < 0.001$

Key: MCI, mild cognitive impairment; MMSE, Mini-Mental State Examination; CDR, clinical dementia rating; SEM, standard error of the mean.

dementia), the mini-mental state exam (MMSE), and global clinical dementia rating (CDR). CDR scores are 0 for normal, 0.5 for questionable dementia, 1 for mild dementia, 2 and 3 for moderate to severe dementia[22]. All participants provided written informed consent and no formal psychological tests or assessments were used to determine whether participants were able to provide written informed consent. The consent procedure and data acquisition procedure of this study were approved by the Institutional Review Board (IRB) of the Korea Centers for Disease Control and Prevention (KCDC) with approval numbers (2016-02-22-P-A, 2017-05-05-P-A). All experiments were performed in accordance with relevant guidelines and regulations.

## Cell cultures

Human umbilical vein endothelial cells (HUVECs) (Lonza, Walkersville, MD, USA) were cultured on 0.2% gelatin-coated wells in Endothelial Growth Medium-2 (EGM-2) media (Lonza) with 2% fetal bovine serum (FBS) at 37˚C in a humidified incubator with 5% $CO_2$, as previously described passages[23] 6–9 were used for experimentation. Primary human brain microvascular endothelial cells (HBMECs) were from Cell systems (Kirkland, WA, USA) and maintained in CSC complete medium with 10% serum and CultureBoost (Cell systems). All primary HBMECs cultures were used between passage 4 and 9. Human iPSC-derived neural progenitor stem cells were obtained from Axol Bioscience (Little Chesterford, UK) and were differentiated to cerebral cortical neurons in approximately 7 days following the recommended manufacturer's protocol.

## Antibodies and Reagents

The following primary antibodies were used: anti-Notch1 intracellular domain (NICD) (Cell Signaling Technology, MA, USA), anti-Notch1 extracellular domain (NEXT) (Thermo Fisher Scientific, MA, USA), anti-Hey-1 (GeneTex, CA, USA), anti-Actin (TransGen, Beijing, China). Synthetic amyloid-beta peptides 1–40 ($A\beta_{1-40}$) were purchased from Invitrogen (Invitrogen, CA, USA) and dissolved in hexafluoreisopropanol (HFIP) (Sigma, MO, USA) for 2 h at room temperature, and lyophilized peptide was dissolved in dimethylsulfoxide (DMSO).

## ELISA measurements

The cell-free plasma samples were stored in aliquoted and stored at -80 ºC until assayed collectively by an investigator who was blinded to the patient assignment. Enzyme-linked

immunosorbent assay (ELISA) was used to measure the human Notch1 level according to the manufacturer's instructions (USCN, Wuhan, China).

## Western blotting

Cells and mouse brains were collected and homogenized in radio-immunoprecipitation assay buffer (RIPA buffer; 20 mM Tris, pH 7.4, 150 mM NaCl, 1 mM $Na_3VO_4$, 10 mM NaF, 1 mM EDTA, 1 mM EGTA, 0.2 mM PMSF, 1% Triton X-100, 0.1% SDS, 0.5% deoxycholate), protein concentration were determined using a Bradford protein assay following the manufacturer's instruction. Bolt 4~12% Bis-Tris gradient gels were used for SDS-PAGE in MES SDS buffer (Life technology, NY, USA). The primary antibodies were diluted in PBS with 5% nonfat dry milk and 0.1% Tween 20 as follows: anti-NICD (1:1,000), anti- NEXT (1:1,000), anti- Hey-1 (1:1,000), and anti-Actin (1:10,000).

## Quantitative real-time polymerase chain reaction (qRT-PCR)

qRT-PCR was performed using SYBR Green PCR core reagent, in a two-step RT-PCR protocol according to the manufacturer's instructions (Applied Biosystems, Warrington, UK). Initial denaturation at 95 ℃ for 10 min was followed by 40 amplification cycles of 95 ℃ for 15 seconds and 58 ℃ for 1 min. The primer sequences are as follows; NOTCH1 sense 5' – `GA GGCGTGGCAGACTATGC -3'` and antisense 5' – `CTTGTACTCCGTCAGCGTGA -3'`; ADAM10 sense 5' – `TGGCCAACCTATTTGTGGAA -3'` and antisense 5' – `CCTCTGGT TGATTTGCATCG -3'`; GAPDH sense 5' – `CAGCCTCAAGATCATCAGCA -3'` and antisense 5' – `TGTGGTCATGAGTCCTTCCA -3'`. GAPDH was used as an internal normalizer. PCR reactions were performed using ABI Prism 7900 SDS (Applied Biosystems, Warrington, UK). Data were analyzed using ΔΔCt method. ΔCt is the difference in the Ct values of the test gene (in each sample assayed) and GAPDH gene, while ΔΔCt represented the difference between the paired samples. All the experiments were performed in triplicates.

## Statistical analysis

Data are presented as the mean ± standard error of the mean (SEM). To analyze demographic and plasma levels of target proteins between dementia, MCI, and control groups, Kruskal-Wallis test was performed followed by Mann-Whitney U-tests. Correlation between factors was analyzed by Spearman's method. Statistical analyses were performed using SPSS 12.0 (IBM, NY, USA). Values of $p < 0.05$ were considered statistically significant. Demographic and clinical differences between control, MCI and dementia group were tested using the Pearson's Chi-squared test, Fisher's exact test and Kruskal-Wallis test.

# Results

## Correlations of sNotch1 levels in dementia patients and healthy subjects

Participant's characteristics are depicted in Table 1. Patients with dementia were older compared to control subjects. The mean age of the control, dementia subjects, and subjects with mild cognitive impairment (MCI) were 72.34 ± 0.37 years, 75.1 ± 0.68 years, and 73.72 ± 0.52 years respectively. The percentage of women amongst control, dementia, and MCI subjects were 57%, 76%, and 60% respectively. Patients with dementia were less educated compared to the control subjects. The overall MMSE scores were lower in patients with dementia, while it was in the normal range in MCI and healthy control subjects.

For analyzing sNotch1 concentration in the plasma, we established a commercial sensitive ELISA method (Fig 1A). Our results showed that the sNotch1 concentration in the plasma was

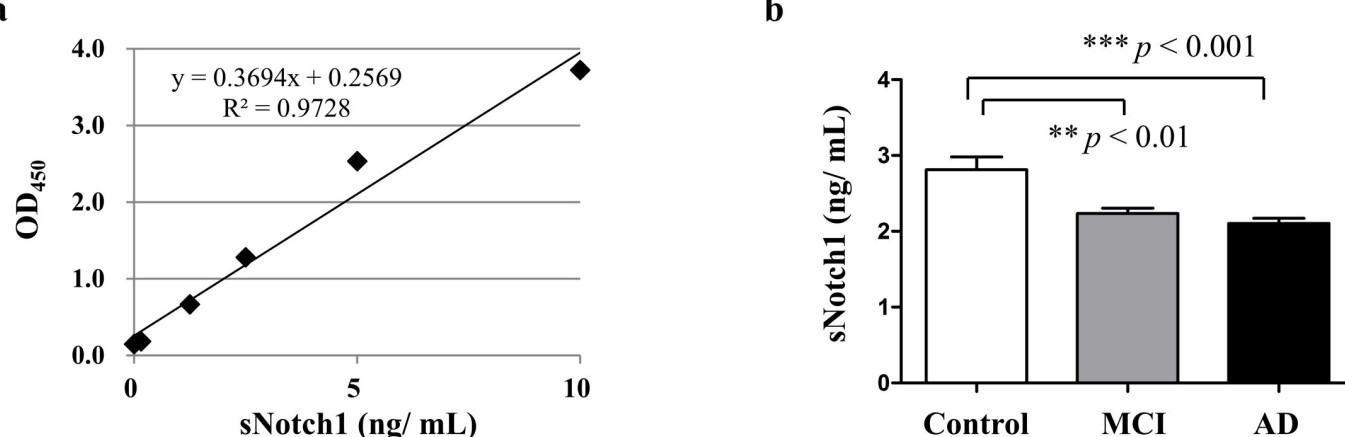

**Fig 1. Comparison of the plasma levels of sNotch1.** (a) Standard curve of the human Notch1 ELISA kit. (b) Plasma sNotch1 concentration was measured by ELISA. The differences in the relative amounts of sNotch1 were compared between control, MCI, and AD by means of Mann-Whitney's U-test within different groups.

different between the three groups ($p < 0.05$; Kruskal-Wallis test). It was lower in subjects with MCI (2.23 ± 0.07 ng/mL) and dementia (2.1 ± 0.07 ng/mL), as compared to the control subjects (2.81 ± 0.17 ng/mL) ($p = 0.015$; Mann-Whitney U-tests) (Table 1). Quantitatively, sNotch1 levels in the plasma samples of dementia patients were 25.27% lower as compared to control subjects (Fig 1B).

Table 2 indicates that there is a positive correlation between the plasma level of sNotch1 and the results of the MMSE assessments in the entire study population. In contrast, the CDR assessment is negatively correlated with the plasma level of sNotch1. No correlation is observed between sNotch1 levels and the age or the gender of the subjects in any of the studied groups; however, it was associated with the participant's level of education.

## Notch1 is increased in endothelial cells by A*β* *in vitro*

To explore the molecular mechanisms regulating sNotch1 level in AD, we investigated sNotch1 levels in endothelial cells (ECs). We examined whether Aβ could modulate sNotch1 levels, which could possibly explain its altered expression in AD patients. ECs were treated with $Aβ_{1−40}$ peptides for 8 h or 24 h, and relative mRNA expression of target genes was measured by quantitative real-time PCR. *Notch1* mRNA level was significantly increased after Aβ treatment for 8 h in human umbilical vein endothelial cells (HUVECs) (Fig 2A) as well as in human brain microvascular endothelial cells (HBMECs) (Fig 3A). Further, we examined the

**Table 2. Correlation between plasma Notch1 and clinical rating scales.**

| Features | Rho | *p* Value |
|---|---|---|
| Ages | 0.016 | 0.781 |
| Education | 0.140 | **0.013** |
| MMSE | 0.134 | **0.018** |
| CDR | -0.117 | **0.038** |

Spearman rank correlation coefficient test was used for assessment of correlations.

Bold values are $p < 0.05$.

Key: MMSE, Mini-Mental State Examination; CDR, clinical dementia rating.

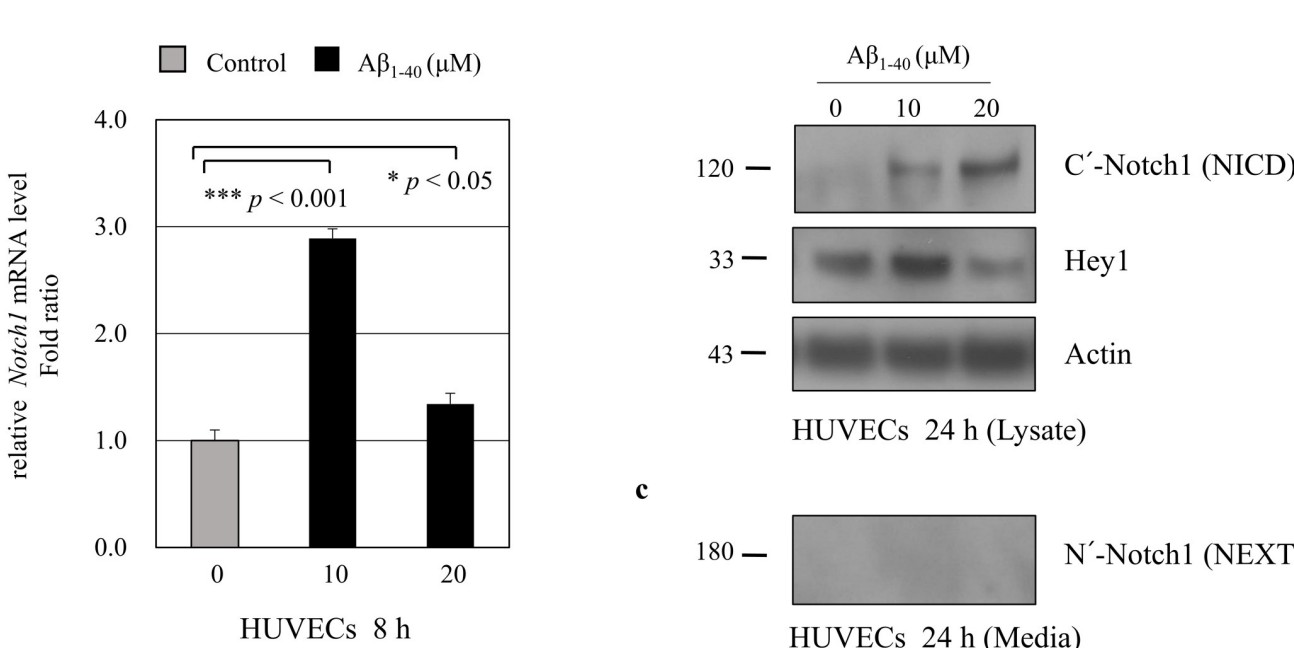

**Fig 2. Modulation of Notch1 expression in HUVECs by amyloid beta (Aβ).** (**a**) HUVECs were treated with 10 μM or 20 μM Aβ$_{1-40}$ peptides for 8 h. Real-time PCR results showing relative mRNA expression levels of *Notch1* (n = 3). (**b**) HUVECs were treated with 10 μM or 20 μM Aβ$_{1-40}$ peptides for 24 h. Notch1 intracellular domain (NICD) and Hey1 protein levels were detected in HUVECs lysate. Treatment with 10 μM Aβ$_{1-40}$ peptides for 24 h significantly increased the Notch1 and Hey1 protein levels (n = 3). Actin was used as a loading control. Secreted soluble Notch1 levels were analyzed by western blotting. Conditioned media were harvested from HUVECs treated with 10 μM or 20 μM Aβ$_{1-40}$ for 24 h. Notch1 extracellular domain (NEXT) protein was not detectable in HUVECs culture media.

Notch1 protein level in ECs. After HUVECs were treated with 10 μM or 20 μM Aβ$_{1-40}$ peptides for 24 h, NICD and Hey1 protein levels were significantly increased (Fig 2B). Consistently, treatment with 5 μM Aβ$_{1-40}$ peptides for 24 h significantly increased the Notch1 and Hey1 protein levels in HBMECs (Fig 3B). These findings suggest that Aβ increases Notch1 mRNA and protein expressions in ECs.

## Modulation of sNotch1 level by Aβ *in vitro*

We next investigated the soluble Notch1 (sNotch1) protein levels in cell culture media using a specific antibody against C-terminus of full-length Notch1, which could detect the secreted sNotch1 extracellular domain (NEXT). After treatment with 5 μM Aβ for 24 h, secreted NEXT protein levels were significantly increased in HBMECs culture media (Fig 3C) but was undetectable in HUVECs culture media (Fig 2C). These findings suggest that Aβ may increase the N-terminal cleavage of full-length Notch1 protein into secreted forms in ECs culture media.

## ADAM10 is increased in endothelial cells by Aβ *in vitro*

To understand how sNotch1 is increased in culture media, we investigated ADAM10 expression levels in endothelial cells (ECs). ECs were treated with Aβ$_{1-40}$ peptides for 8 h or 24 h, and relative mRNA expression of target genes was measured by quantitative real-time PCR. After treatment with 10 μM Aβ$_{1-40}$ peptides, *ADAM10* mRNA level was significantly increased in HUVECs (Fig 4A, * $p < 0.05$) and HBMECs (Fig 4B, ** $p < 0.01$). Increased ADMA10 mRNA expression might affect to the cleavage of Notch1 extracellular domain under the Aβ-treated

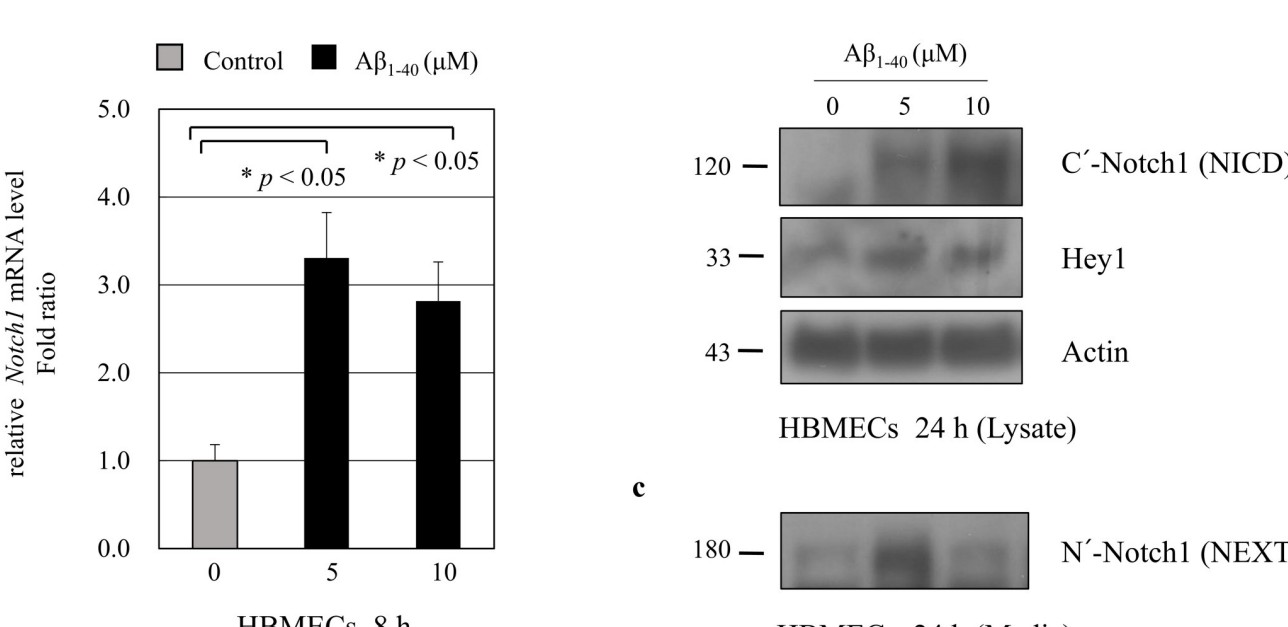

**Fig 3. Modulation of Notch1 expression in HBMECs by amyloid beta (Aβ). (a)** HBMECs were treated with 5 μM or 10 μM Aβ$_{1-40}$ peptides for 8 h. Real-time PCR results showing relative mRNA expression levels of *Notch1* (n = 3). (**b**) HBMECs were treated with 5 μM or 10 μM Aβ$_{1-40}$ peptides for 24 h. Notch1 intracellular domain (NICD) and Hey1 protein levels were detected in HBMECs lysate. Treatment with Aβ$_{1-40}$ peptides for 24 h significantly increased the Notch1 and Hey1 protein levels (n = 3). Actin was used as a loading control. Secreted soluble Notch1 levels were analyzed by western blotting. Conditioned media were harvested from HBMECs treated with 5 μM or 10 μM Aβ$_{1-40}$ for 24 h. After treatment with 5 μM Aβ peptides, Notch1 extracellular domain (NEXT) protein level was significantly increased in HBMECs culture media.

condition in ECs. So, it could be explain that secreted sNotch1 extracellular domain (NEXT) is increased in HBMECs culture media.

### Aβ increases Notch1 expression in human iPSC-derived neuronal cells

We next examined *Notch1* mRNA expression in human iPSC-derived neuronal cells by quantitative real-time PCR. *Notch1* mRNA levels were measured in human iPSC-derived neuronal cells from an AD patient and a healthy control subject (n = 3). Human iPSC cells were differentiated into neurons using commercial neuronal differentiation media. As we expected, *Notch1* mRNA expression was significantly increased in human iPSC-derived neuronal cells from an AD patient compared to a healthy control subject with $p < 0.05$ (Fig 5A). We also measured the *Hey1* and *Hes5* mRNA levels in iPSC-derived AD neurons and healthy neurons (n = 3, respectively). *Hey1* and *Hes5* mRNA expression were significantly increased in human iPSC-derived neuronal cells from an AD patient with both $p < 0.01$ (Fig 5B and 5C). These results suggest that Notch1 levels are increased in patients with AD.

### Discussion

Alzheimer's disease (AD) and vascular dementia (VAD) are the two main types of dementia affecting approximately 70% and 15%, respectively, of all demented patients[24]. Recent neuropathological studies have demonstrated that patients with AD have concomitant cerebrovascular pathology[25]. Notch signaling is involved in a number of cellular processes such as cell-cell interactions, crosstalk with neighboring cells, vasculogenesis, angiogenesis,

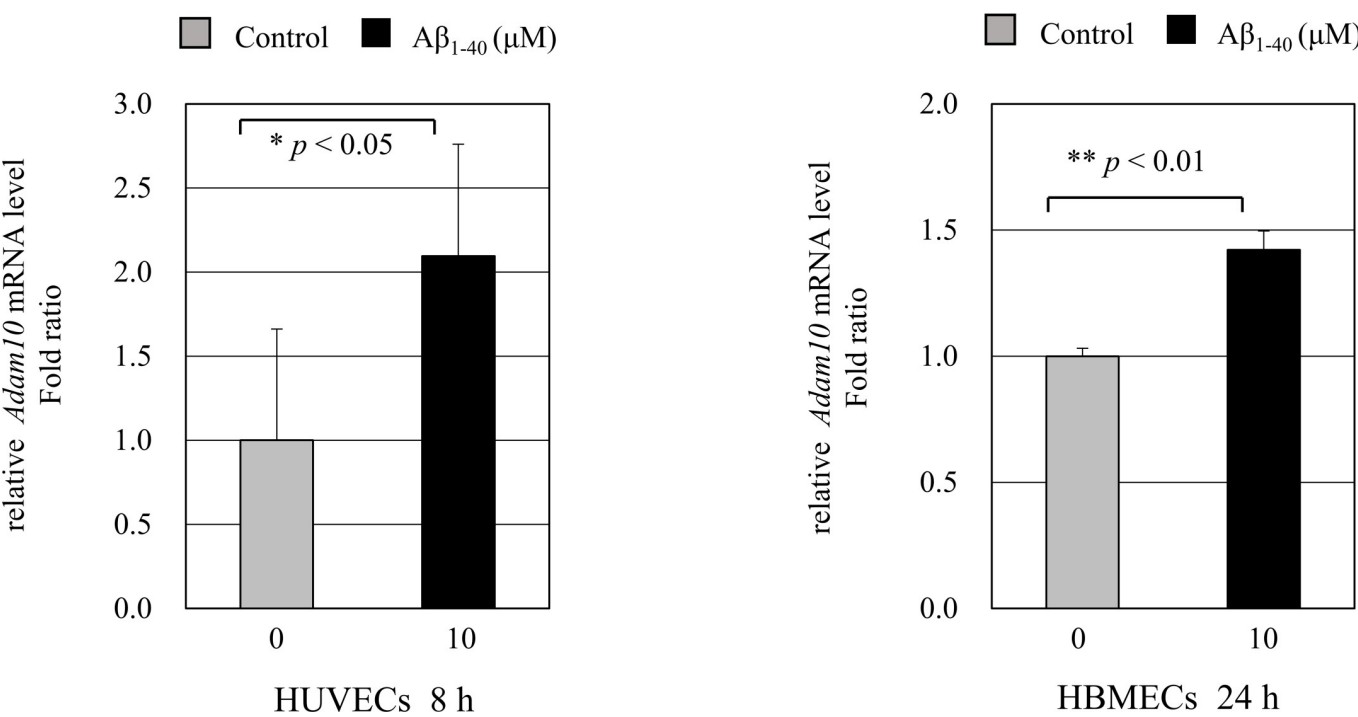

**Fig 4. *ADAM10* mRNA expression in ECs by amyloid beta (Aβ).** HUVECs (a) and HBMECs (b) were treated with 10 μM Aβ$_{1-40}$ peptides for 8 h or 24 h. Real-time PCR results showing relative mRNA expression levels of *ADAM10* (n = 3). Treatment with 10 μM Aβ$_{1-40}$ peptides significantly increased the *ADAM10* mRNA expression levels (* $p < 0.05$ and ** $p < 0.001$, respectively).

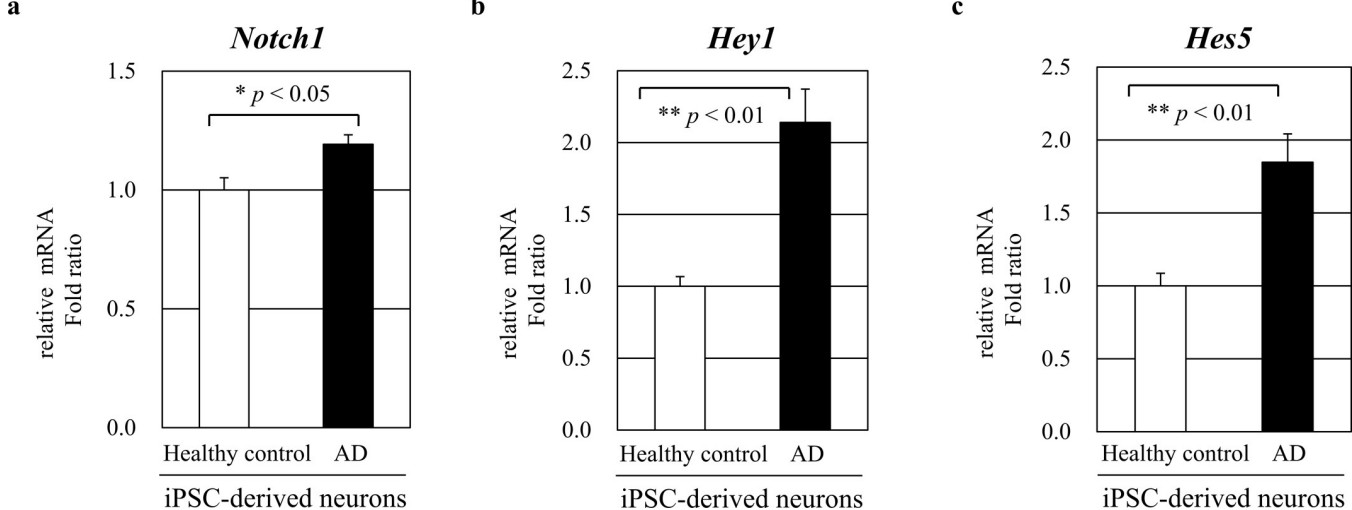

**Fig 5. *Notch1* mRNA expression in human iPSC-derived neuronal cells.** (**a**) Relative *Notch1* mRNA expression levels were measured in human iPSC-derived neuronal cells from an AD patient and a healthy control subject (n = 3). Human iPSC cells were differentiated to neurons by treating with neuronal differentiation media. *Notch1* mRNA expression was significantly increased in human iPSC-derived neuronal cells from an AD patient (* $p < 0.05$). (b-c) Relative *Hey1* and *Hes5* mRNA expression levels were measured (n = 3, respectively) and significantly increased in human iPSC-derived neuronal cells from an AD patient (both ** $p < 0.01$).

permeability, and survival. Although Notch signaling cascade responds to microenvironmental conditions through cell-cell interactions, several other stimuli including aging, shear stress, inflammation, hypoxia, and hyperglycemia are known to activate this pathway[4, 26–29].

In endothelial cells (ECs), Notch signaling has been reported to modulate angiogenesis through the repression of VEGFR2[30]. Angiogenesis and vascular dysfunction may be involved in neurodegeneration[31]. VEGF levels are increased in tissues obtained from the patients with AD[32] and VEGF/Notch signaling cascade may play an important role in the pathology of AD including BBB leakage. Notch1 is proteolytically cleaved by γ-secretase, releasing the NICD into the cytoplasm, which then translocates to the nucleus and regulates expressions of several genes including *Hey1* and *Hes5*[33]. Previous studies have reported that Notch1 is upregulated in the brain of AD patients[34, 35]. Notch1 plays an important role in the maintenance of neuronal stem cells[36], regulating neurogenesis[37], and activating stem cells in response to injury for increased neurogenic demand[38]. Activation of Notch1 signaling induced by cerebral ischemia results in a protracted nuclear factor-κB (NF-κB)-driven microglia-mediated neuroinflammation and worsens ischemic brain damage and functional outcome[39]. Activated microglial cells are detected in peri-infarct areas in clinical stroke and though to orchestrate neuronal damages in the penumbra[10]. Recent studies raise the possibility of the links between the Notch signaling pathway and diverse pathological disorders including adaptive and innate immune responses, multiple sclerosis, inflammatory demyelinating disease, which is providing Notch as a novel prospective target for the treatment of neuroinflammation-related degenerative disorders.

Notch signaling is also important for normal adult brain function and is implicated in various neurological diseases. However, the role of Notch signaling in fine-tuning neurogenesis and neurodegenerative pathology is not fully understood. Studies showed a link between Notch and neurological disorders including Alagille syndrome, cerebral autosomal dominant arteriopathy with subcortical infarcts and leukoencephalopathy (CADASIL) syndrome, mental retardation, and certain types of schizophrenia[40]. In addition, Notch1 is overexpressed in the neurogenic and non-neurogenic regions of the brain in sporadic Alzheimer's disease and adults with Down syndrome[41]. Consistent with these findings, we found that Notch1 expression was increased in Aβ-treated ECs. These results indicate that the levels of Notch1 and sNotch1 might be associated with dysregulated proteostasis in neurodegenerative disorders such as AD. Mutations in *Notch2* gene induces Down syndrome and *Notch3* mutations lead to cardiovascular disorder CADASIL that causes stroke and vascular dementia with degenerative changes in the vascular smooth muscles. Several familial AD mutations are known to be associated with diminished Notch activity but the mechanisms remain elusive[38].

Recent studies showed that chronic hippocampal expression of NICD induces vascular thickening with the accumulation of amyloid beta (Aβ) which exacerbates spatial memory deficits in a rat model of early AD. In addition, chronic activation of Notch1 signaling causes impaired blood flow while reducing nutrient delivery and worsening brain function in a McGill-R-Thy1-APP transgenic (Tg) rat model of early AD. These results suggest that chronic activation of Notch1 may accelerate Aβ accumulation and spatial memory deficits in Tg rodent models of AD[42].

Notch signaling has a critical roles in arterial formation and maturation. Dll4 and Jagged1 are important Notch1 ligands and were shown to have opposing effects in developmental angiogenesis. But the role of Jagged1/Notch signaling remains incompletely understood[43]. Notch signaling is associated with several pathological conditions and recent study shows that Notch signaling foster macrophage maturation during ischemia causing inflammatory responses[44]. Further studies will be required to examine Jagged1 and Dll4 to understand the role of Notch signaling in AD.

In our previous study, we showed the plasma levels of soluble VEGFR2 (sVEGFR2) in association with AD-related cognitive decline[8] and overexpression of NICD decreased the *VEGFR2* mRNA levels in HUVECs[8]. Since recent study has reported that Notch1 is increased in AD[12], we further determined the alteration of the plasma levels of Notch1 in AD with the population as described in detail previous[8, 16].

Here we investigated Notch1 expression in human AD patients. We found that NICD and secreted soluble NEXT protein levels are significantly increased in Aβ-treated ECs *in vitro*. *Notch1* mRNA expression is also significantly increased in human iPSC-derived neuronal cells. These results might implicate an increase in the Notch1 deposition in the amyloid plaques within the brain tissue of AD patients. Although Notch1 expression was induced by Aβ in ECs, the plasma levels of sNotch1 were significantly reduced in patients with AD. Our study provides a possible explanation that increased full-length and truncated Notch1 proteins are localized at the amyloid plaques, thereby reducing the sNotch1 level in the plasma. Decrease in plasma sNotch1 level consequently leads to reduced full-length and truncated Notch1 protein in cerebrospinal fluid (CSF) of AD patients[12]. The present study demonstrates that the plasma level of Notch1 correlates with cognitive decline in patients with dementia. We found that plasma levels of Notch1 were significantly lower in patients with AD than in patients with MCI or healthy control subjects. Accumulation of Notch1 and colocalization with the amyloid-beta plaques in AD patient's brain might explain the reduced levels of plasma sNotch1 in these patients. Further studies are required to understand the mechanism of decreased plasma sNotch1 levels in AD, which will be helpful in deciphering the role of Notch1 signaling in neuropathological conditions.

In conclusion, our results show that plasma sNotch1 levels are decreased in patients with AD. We suggest that alteration in sNotch1 level might be associated with those at risk for Alzheimer's disease.

## Supporting information

**S1 Fig. Original uncropped and unadjusted full blots of Fig 2B and 2C.**
(PDF)

**S2 Fig. Original uncropped and unadjusted full blots of Fig 3B and 3C.**
(PDF)

## Acknowledgments

All participants were collected and clinically evaluated by Moon Ho Park and Changsu Han from Korea University Ansan Hospital.

## Author Contributions

**Conceptualization:** Young Ho Koh.

**Data curation:** Sun-Jung Cho.

**Formal analysis:** Sun-Jung Cho, Sang-Moon Yun, Chulman Jo.

**Investigation:** Sun-Jung Cho.

**Methodology:** Sun-Jung Cho.

**Resources:** Moon Ho Park, Changsu Han.

**Software:** Jihyun Jeong, Young Ho Koh.

**Supervision:** Young Ho Koh.

**Writing – original draft:** Sun-Jung Cho.

**Writing – review & editing:** Sang-Moon Yun, Chulman Jo, Young Ho Koh.

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
