## [Decision Letter · Decision Letter 0]

8 Aug 2019

PONE-D-19-18002

Altered expression of Notch1 in Alzheimer's disease

PLOS ONE

Dear Dr. Koh,

Thank you for submitting your manuscript to PLOS ONE. After careful consideration, we feel that it has merit but does not fully meet PLOS ONE’s publication criteria as it currently stands. Therefore, we invite you to submit a revised version of the manuscript that addresses the points raised during the review process.

We would appreciate receiving your revised manuscript by Sep 22 2019 11:59PM. To enhance the reproducibility of your results, we recommend that if applicable you deposit your laboratory protocols in protocols.io, where a protocol can be assigned its own identifier (DOI) such that it can be cited independently in the future. For instructions see: http://journals.plos.org/plosone/s/submission-guidelines#loc-laboratory-protocols

We look forward to receiving your revised manuscript.

Kind regards,

Dong-Gyu Jo, Ph.D

Academic Editor

PLOS ONE

Journal Requirements:

1. Please describe in your methods section how capacity to provide consent was determined for the participants in this study. Please also state whether your ethics committee or IRB approved this consent procedure.

2. Our internal editors have looked over your manuscript and determined that it may be within the scope of our Early Diagnosis and Treatment of Alzheimer's Disease Call for Papers. This collection of papers is headed by a team of Guest Editors for PLOS ONE: Michael Weiner, Roberta Brinton, Jussi Tohka and Yona Levites. With this Collection we hope to bring together researchers working on a wide range of disciplines, from molecular and preclinical work, through to patient-centered studies, including clinical trials.   Additional information can be found on our announcement page: https://collections.plos.org/s/alzheimersdisease. If you would like your manuscript to be considered for this collection, please let us know in your cover letter and we will ensure that your paper is treated as if you were responding to this call. If you would prefer to remove your manuscript from collection consideration, please specify this in the cover letter.

Reviewers' comments:

Reviewer's Responses to Questions

**Comments to the Author**

1. Is the manuscript technically sound, and do the data support the conclusions?

Reviewer #1: Partly

Reviewer #2: Partly

2. Has the statistical analysis been performed appropriately and rigorously? 

Reviewer #1: Yes

Reviewer #2: Yes

3. Have the authors made all data underlying the findings in their manuscript fully available?

Reviewer #1: Yes

Reviewer #2: Yes

4. Is the manuscript presented in an intelligible fashion and written in standard English?

Reviewer #1: No

Reviewer #2: Yes

5. Review Comments to the Author

Reviewer #1: 1. Table1: They also should demonstrate the specificity of the antibody.

2. Table1: The bar plot would help to understand the variations of the data.

3. Figure 3: The authors described that mRNA levels of Notch1 are increased iPSC-derived AD neurons. Do you think it is Notch1 specific? The authors also should measure at least one more gene in these samples, hey-1 or hes-5.

4. Before resubmitting, I would recommend having the manuscript reviewed by a native English speaker.

5. The manuscript should format according to the jorunal instruction. some parts including reference part is not well formatted.

Reviewer #2: In this paper, the authors show that Notch1 signaling is modulated by pathological condition such as Alzheimer’s disease. The manuscript is well-written, but the current data is missing some points.

1. As mentioned by the authors, it is well known that Notch1 extra-cellular domain is cleaved by metalloprotease ADAM10. To understand how sNotch1 is increased in culture media need to confirm the ADAM 10 expression.

2. Since Notch1 ligands, Dll4 and Jag1, have opposite effects and is required to activate Notch1 signaling in endothelial cell differentiation in artery. Investigation of these protein will give more information about activation of Notch1.

3. Notch1 signaling is activated by not only development stage but also pathological condition such as ischemic stroke, neuronal death, and followed inflammatory responses. Please, cite and discuss those related previous findings which will be helpful to understand role of Notch1 signaling in neurodegenerative diseases.

6. PLOS authors have the option to publish the peer review history of their article (what does this mean?). If published, this will include your full peer review and any attached files.

Reviewer #1: No

Reviewer #2: No

---

## [Author Response · Author response to Decision Letter 0]

20 Sep 2019

Reviewer #1: I have incorporated all of your suggestions into my revision. Thank you.

Reviewer #2: I have incorporated all of your suggestions into my revision. Thank you.

---

## [Editor Report · Decision Letter 1]

25 Oct 2019

Altered expression of Notch1 in Alzheimer's disease

PONE-D-19-18002R1

Dear Dr. Koh,

We are pleased to inform you that your manuscript has been judged scientifically suitable for publication and will be formally accepted for publication once it complies with all outstanding technical requirements.

With kind regards,

Dong-Gyu Jo, Ph.D

Academic Editor

PLOS ONE
---

## [Editor Report · Acceptance letter]

18 Nov 2019

PONE-D-19-18002R1 

Altered expression of Notch1 in Alzheimer's disease 

Dear Dr. Koh:

I am pleased to inform you that your manuscript has been deemed suitable for publication in PLOS ONE. Congratulations! Your manuscript is now with our production department. 

With kind regards,

on behalf of

Dr. Dong-Gyu Jo 

Academic Editor

PLOS ONE